# Fabrication of triboelectric polymer films via repeated rheological forging for ultrahigh surface charge density

Zhaoqi Liu[1,2,3,7], Yunzhi Huang[2,7], Yuxiang Shi [1,3], Xinglin Tao[1,3], Hezhi He[2], Feida Chen[4], Zhao-Xia Huang [2✉], Zhong Lin Wang [1,3,5], Xiangyu Chen [1,3✉] & Jin-Ping Qu [2,6✉]

Triboelectric polymer with high charge density is the foundation to promote the wide range of applications of triboelectric nanogenerators. This work develops a method to produce triboelectric polymer based on repeated rheological forging. The fluorinated ethylene propylene film fabricated by repeated forging method not only has excellent mechanical properties and good transmittance, but also can maintain an ultrahigh tribo-charge density. Based on the film with a thickness of 30 μm, the output charge density from contact-separation nanogenerator reaches 352 μC·m$^{-2}$. Then, the same film is applied for the nanogenerator with air-breakdown mode and a charge density of 510 μC·m$^{-2}$ is further achieved. The repeated forging method can effectively regulate the composition of surface functional groups, the crystallinity, and the dielectric constants of the fluorinated ethylene propylene, leading to the superior capability of triboelectrification. Finally, we summarize the key parameters for elevating the electrification performance on the basis of molecular structure and related fabrication crafts, which can guide the further development of triboelectric polymers.

[1] CAS Center for Excellence in Nanoscience, Beijing Institute of Nanoenergy and Nanosystems, Chinese Academy of Sciences, 100083 Beijing, China. [2] National Engineering Research Center of Novel Equipment for Polymer Processing; Key Laboratory of Polymer Processing Engineering, Ministry of Education; Guangdong Provincial Key Laboratory of Technique and Equipment for Macromolecular Advanced Manufacturing; Department of Mechanical and Automotive Engineering, South China University of Technology, 510641 Guangzhou, China. [3] School of Nanoscience and Technology, University of Chinese Academy of Sciences, 100049 Beijing, China. [4] Department of Nuclear Science & Engineering, Nanjing University of Aeronautics and Astronautics, Nanjing, China. [5] School of Materials Science and Engineering, Georgia Institute of Technology, Atlanta, GA 30332-0245, USA. [6] School of Chemistry and Chemical Engineering, Huazhong University of Science & Technology, 430074 Wuhan, China. [7] These authors contributed equally: Zhaoqi Liu, Yunzhi Huang. ✉email: mehuangzx@scut.edu.cn; chenxiangyu@binn.cas.cn; jpqu@scut.edu.cn

n recent years, increasing demand of clean and sustainable energy has drawn a lot of attention to triboelectric nanogenerators (TENG), which can collect low-frequency and irregular kinetic energy into electricity[1]. In order to realize the real industrialization of TENG, the output power density is the most important parameter that needs to be further enhanced[2]. Therefore, researchers have developed a variety of structures and optimization methods for TENG to amplify the electrostatic energy output, such as free-standing structure[3] and direct current TENG based on air-breakdown effect[4,5], while a series of charge management strategies have also been proposed for better utilizing the electrostatic charges[6,7] generated by TENG, including charge pumping system[8], vacuum/oil protection strategy[9,10] and so on. Accordingly, the calculated charge density from the output end of the TENG system has been greatly improved. However, no matter how to optimize the system structure and the charge utilization strategy, the generated energy of the nanogenerator still originates from the intrinsic surface charge density of the triboelectric material[11,12]. Hence, the triboelectric material is always the foundation of TENG technique, while more efforts should be devoted to the development of triboelectric materials with high charge density and diversified functions, and there is still a lack of a guidance on the fabrication of high-performance triboelectric materials.

Currently, commercial tapes are the most widely used triboelectric polymers for the study of TENG, since they are the most accessible objects on the market possessing the quality of both smooth surface and high electrification capability[13,14]. However, the manufacturing technology for commercial tape is not specially designed for tribo-charge generation and it is highly desirable that some different fabrication methods especially targeting the electrification capability can be proposed for producing triboelectric polymers. Among all these polymer tapes, Fluorinated Ethylene Propylene (FEP) has quite superior electrification capability and its ranking is also at the very negative position of the triboelectric series, which allows it to be widely used as self-powered sensors and energy packages. Even though many modification methods, including surface etching[15–17], surface modification of materials[18,19] and ion implantation[20–22], have

been applied for FEP tapes, the charge density of the FEP tape with a thickness of 30~50 μm still cannot exceed the limits of 250 μC·m⁻². On the other hand, previous studies have proved that functional groups on the molecular chains of polymer have a vital impact on contact electrification[23,24]. Hence, the underlying relationships between the molecular structure of the objects and their macroscopic electrification properties can inspire us with a different approach to develop triboelectric polymers with high charge density[25].

In this paper, we propose a fabrication method to effectively regulate the molecular structure of triboelectric polymer by repeated rheological forging (RRF) process, which can produce scalable triboelectric film with ultrahigh charge density. The FEP film prepared by RRF method (RRF-FEP) maintains excellent mechanical toughness and light transmittance, while its intrinsic surface charge density is 1.46 times of the highest records reported before. The rheological forging with different releasing times can effectively regulate the molecular orientation, the crystallinity and even the dielectric constant of the material. These changes in molecular structure and crystallinity lead to distinct electrification performance before and after corona polarization, suggesting different approaches for boosting the triboelectric performance of electret and non-electret polymers. Triboelectric polymer with high charge density is the foundation of TENG devices and thus, the proposed RRF-FEP in this work can be applied to fabricate various TENG devices. Moreover, we summarize the systematic strategies for designing and producing high-performance electrification films on the basis of the results in this work, which may guide a series of possible breakthroughs in the future study of higher performance triboelectric materials.

## Results

**Fabrication of FEP films by repeated rheological forging.** The production process of different RRF-FEP films is shown in Fig. 1a and Supplementary Movie 1. The FEP materials are firstly molten at 300 °C, and then the molecular chains of FEP are reformed by repeating the pressing-releasing fabrication processes 120 times. After molding, samples are cooled in the air to produce RRF-FEP

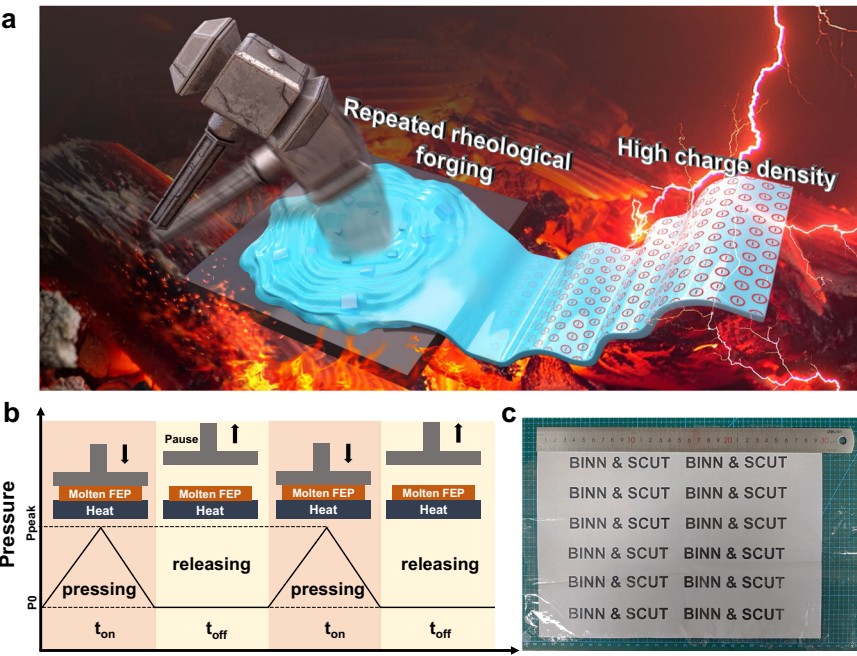

**Fig. 1 Repeated rheological forging technique and molding film. a** Production of high triboelectric charge density films by rheological repeated forging technique. **b** Illustration of Repeated rheological forging technique. **c** Physical map of FEP film produced by RRF process.

films. As shown in Fig. 1b, a production cycle can be divided into two parts: the pressing section and the releasing section. In the pressing section, the molecular chains of FEP are confined and compressed in the mold, resulting in the decrease of the free volume, and the ordered chain packing. Then, in the releasing section, the compressed material starts to recover from such a thermodynamically unfavorable state to a thermodynamically stable state, in which the chains tend to be disordered and entangled. Meanwhile, if we apply different relaxation time to the polymer during the hot pressing, the polymer molecules can be fixed at a confined state. This relaxation time ($t_{off}$) is the key parameter to the formation of polymers with different structures, and different $t_{off}$ may be related to the different spatial configurations. Therefore, four different RRF-FEP films are prepared by changing the $t_{off}$ in the molding process (RRF-FEP1-4), and the detailed parameters are shown in Supplementary Table S1. As shown in Fig. 1c, the prepared large-size (20 cm × 30 cm) RRF-FEP film shows excellent optical transmission. Among all the easily accessible polymer tapes on the market, FEP is usually believed to have the highest tribo-charge density[26]. Hence, the FEP is selected for demonstrating the rheological forging technique, while a similar forging technique can also be applied for other polymers.

**Characterization of molding FEP.** The properties of the prepared films are compared with those of commercial FEP (Shanghai Weitelang Industrial Co., Ltd.), which is the best FEP that can be accessible by us. The surface morphology of the film is observed by atomic force microscope (AFM) in Fig. 2a, where the average roughness is also calculated. In our experiment, we chose

the pressure of 50 MPa and the surface roughness of RRF-FEP and the commercial FEP can be maintained all at similar levels, which proves that the forging method does not change the flatness of the film surface. It is worth noting that if the applied pressure is very low, the flatness may have some difference. However, the pressure provided by our machine is usually larger than the pressure threshold of the imprinting and the influence of pressure on the surface roughness is negligible. This can also be confirmed by scanning electron microscopy (SEM) of films in Supplementary Fig. S1. The tensile strain-stress curves (see in Fig. 2b) of RRF-FEP and commercial FEP are tested by a universal testing machine to characterize the mechanical properties of materials. Compared with the commercial FEP, the mechanical properties of RRF-FEP films have been significantly improved. For the RRF-FEP3, the elongation at the break of the material increases from 349% to 515%, and the tensile strength increases from 9 MPa to 21 MPa. RRF-FEP3 has the largest tensile modulus of 161.6 MPa, which is nearly twice that of commercial FEP. The repeated forging process makes the molecular chains to be at a highly confined state so that the mechanical properties of the material have been reinforced. As further shown in Supplementary Fig. S2, the surface elastic modulus of commercial FEP and RRF-FEP3 is compared by AFM. It has been found that the surface elastic modulus of the material also increased from 200 MPa to 600 MPa after RRF processing. Moreover, dynamic mechanical analysis (DMA) has also been conducted to provide more information about the mechanical properties of RRF-FEP films in comparison with commercial ones. As shown in Supplementary Fig. S3, the molding method did not change the glass transition temperature of FEP, but in the room temperature

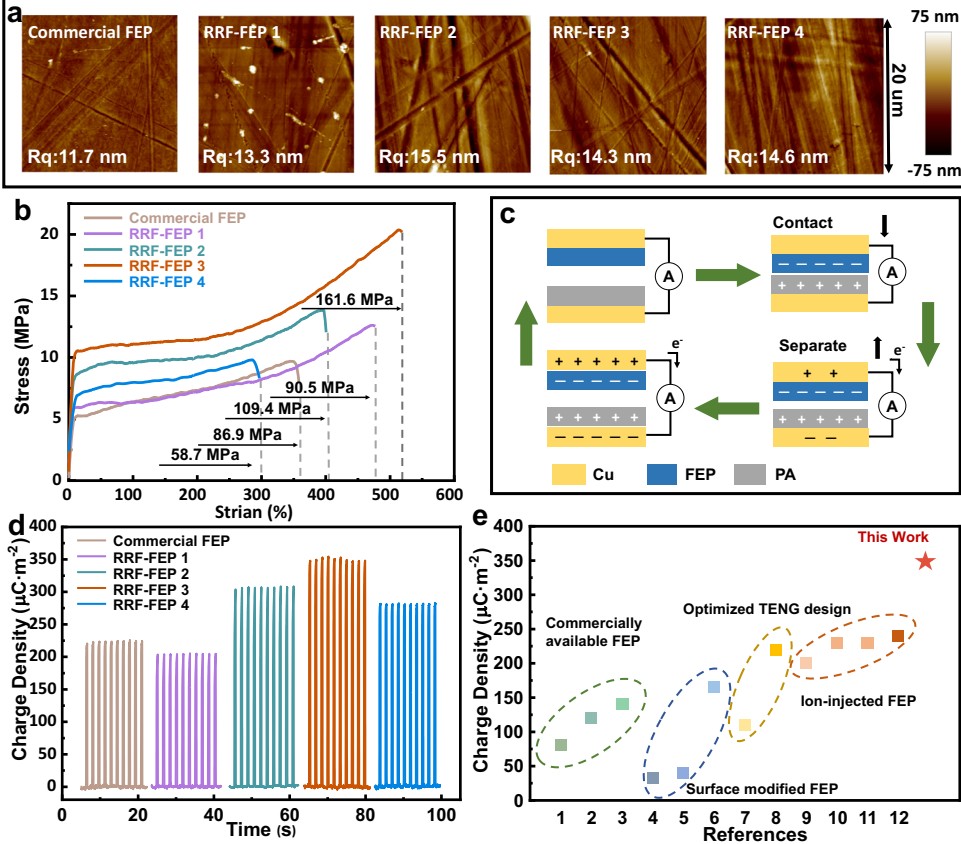

**Fig. 2 High performance demonstration of RRF-FEP compared with commercial FEP. a** The surface morphology of commercial FEP and RRF-FEP observed by atomic force microscope (AFM) (**b**) Stress vs strain curves of each sample during tensile deformation. **c** Schematic diagram of measurement method in experiment with FEP samples. **d** Maximum saturated charge density between FEP and PA measured by contact-separation TENG. **e** Saturated charge density comparison in the previous works of the TENG. Source data are provided as a Source Data file.

range, RRF-FEP3 shows a higher storage modulus, indicating a higher strength. To provide insight into the thermal degradation of the polymer as it is subjected to an increasing number of forging cycles, Thermogravimetric analysis (TGA) results of commercial FEP, raw FEP powders (0 cycle) and RRF-FEP3 (60 cycles, 120 cycles and 180 cycles) are provided as below (Supplementary Fig. S4). The results revealed that all samples show similar thermal degradation behaviors with temperature from 25 °C to 800 °C, indicating no obvious chain degradation occurred during the repeated forging process. As can be seen in Fig. 2c, we select vertical contact-separation TENG (CS-TENG) for the experiments to test the intrinsic charge density of different FEP films. FEP and Polyamide (PA) are the targeting materials, while the positive and negative charges are respectively generated during the contact. Then, during the separation process, the Cu electrodes behind PA and FEP induce opposite charges based on the electrical induction effect and the electrometer can detect the movement of the charges between the two electrodes. Before measurement, a corona polarization is applied on each FEP film with the same polarization parameters[27,28], which can help to achieve a maximum surface charge density. Each film is measured ten times repeatedly to ensure the repeatability of the experimental results, and the result is shown in Fig. 2d. For RRF-FEP3, its surface charge density reaches 352 $\mu C \cdot m^{-2}$, which proves that RRF can effectively improve the triboelectrification properties of FEP. In previous work, many approaches have been used to improve the charge density of FEP. For commercially available original FEP[18,19,29,30], the charge density is usually not high without surface modification, approximately 140 $\mu C \cdot m^{-2}$. Surface etching[15–17] is a physical modification method to increase the charge density of FEP, where the micro-nanopattern established on the surface by the inductively coupled plasma etching can effectively increase the contact area of TENG. However, this method is still limited by the intrinsic charge density of the material and the obtained surface charge density cannot exceed 200 $\mu C \cdot m^{-2}$. Optimizing the structure[14,31] of TENG does not change the triboelectrification capacity of FEP from the material itself. Meanwhile, corona polarization[5,20–22] is also one of the most effective methods for improving the charge density, which can provide a charge density of 240 $\mu C \cdot m^{-2}$ for commercial FEP. Nevertheless, corona polarization can only work effectively with electret materials and the charge density is also decided by the electret characteristics, which means the material design is still the key element. As marked in Fig. 2e and Supplementary Table S2, this work has achieved the highest charge density for FEP (with the thickness between 30–50 $\mu m$) based on CS-TENG mode, and this value is 1.46 times of the highest previous records.

In order to further study the underlying mechanism of the performance enhancement of RRF-FEP, in-depth characterizations of materials properties have been performed. Figure 3a shows the charge density of FEP films in contact with PA without high voltage polarization, measured by CS-TENG. The charge density of commercial FEP can reach 83.8 $\mu C \cdot m^{-2}$, while the best RRF-FEP can only reach 30.5 $\mu C \cdot m^{-2}$. The output charge density without polarization is related to the electron-withdrawing capability of functional groups of the polymer. Hence, we assume that the molecular structure of commercial FEP should be highly oriented (see Fig. 3b), where the -$CF_3$ group with stronger electron-withdrawing capability is facing the surface region. On contrary, the -$CF_3$ groups in the RRF-FEP should be non-oriented (see Fig. 3c), which may suppress the possibility of electron transfer during the contact. It is important to note that we cannot find the detailed fabrication process of commercial FEP and our speculation is only based on the results of material characteristics. After high voltage corona poling, the electrification performance of all FEP films has been improved greatly, and

the high voltage polarization device is shown in Fig. 3d. The high voltage polarization may have two major contributions. The high voltage between the needle electrode and the Cu electrode deflects the dipole moment of the molecules in FEP. Then, the needle electrode ionizes the air above the FEP and injects a negative charge into the FEP surface. The surface charge on the FEP is saturated by high voltage polarization, and FEP can maintain this charge for a long time as an electret. Figure 3e shows the change of charge density before and after polarization. Here, the charge density of RRF-FEP after polarization has been greatly improved, especially for RRF-FEP3, whose charge density after polarization is 17.5 times that before polarization. It is proved that RRF-FEP2 has the strongest electron-withdrawing ability, while RRF-FEP3 has the strongest charge storage capability.

**Mechanisms of enhancement of RRF process.** The molecular structures of commercial FEP and RRF-FEPs are given by a Fourier transform infrared (ATR-FTIR) spectrum (see Fig. 4a), the -$CF = CF_2$ stretching at 1647 $cm^{-1}$ appears in RRF-FEP1-3, and the results agree with the result of C1s spectrum measured by X-ray photoelectron spectroscopy (XPS) (see Supplementary Fig. S5 and Table S3). Increasing number of -$CF = CF_2$ groups is considered to be evidence of that repeated forging is capable to induce more end groups in the molecular chain. Usually, the -$CF = CF_2$ group is the end part of the molecular chain[32]. Thus, we infer that, at least in the surface region, a certain amount of polymer chains is possibly cleaved during the repeated pressing process and a series of short chains are generated, resulting in more end groups (-$CF = CF_2$) observed in the experiments. As mentioned before, $t_{off}$ is the key factor of this forming method, the result shows that the intensity of end groups first increases and then decreases along with longer $t_{off}$ during RRF processing and the intensity reaches the peak when $t_{off}$ is 0.95 s. In previous studies[26], it has been proved that the double bond in the molecular chain affects triboelectrification due to the conjugate effect. Furthermore, the molecular electrostatic potentials of the chain element of FEP with or without -$CF = CF_2$ are calculated by DFT, and the results are shown in Supplementary Fig. S6. The blue area is the positive potential area, corresponding to the electron poor area. It is found that the -$CF = CF_2$ bond can lead to an electron poor region at the end group (the blue area in the chain element). This result explains the charge densities order of RRF-FEP under four different molding parameters before polarization in Fig. 3a. In the case of $t_{off}$ is 0.95 s, RRF-FEP2 has the most -$CF = CF_2$ groups. Moreover, there is a larger proportion of -$CF_3$ groups on its surface, and RRF-FEP2 has the best triboelectrification performance among four RRF-FEPs.

Figure 4b and Supplementary Fig. S5 show the molecular structure of FEP surface measured by XPS. Since the detection depth of XPS is less than 10 nm, it can further verify the change of the surface functional group of FEP. Compared with RRF-FEP, commercial FEP has more proportion of -$CF_3$ and less -$CF_2$ on the surface, leading to the high triboelectric performance before corona polarization. As shown in Fig. 4c, we used 2d-WAXD (2d-Wide-angle X-ray diffraction) to observe the crystalline information of all materials. As suggested by isotropic 2d-WAXD rings, the FEP obtained by RRF has no significant orientations, while the commercial FEP has a clear orientation which is believed due to its fabrication process. The results of XPS and 2d-WAXD confirm the structures of commercial FEP and RRF-FEP in Fig. 3b, c, where the molecular structure of commercial FEP is oriented with the -CF3 groups facing the surface and the -CF3 groups in the RRF-FEP is non-oriented. Commercial FEP has great orientation leading to there being more -$CF_3$ functional groups on the surface resulting in its high charge density before it is polarized.

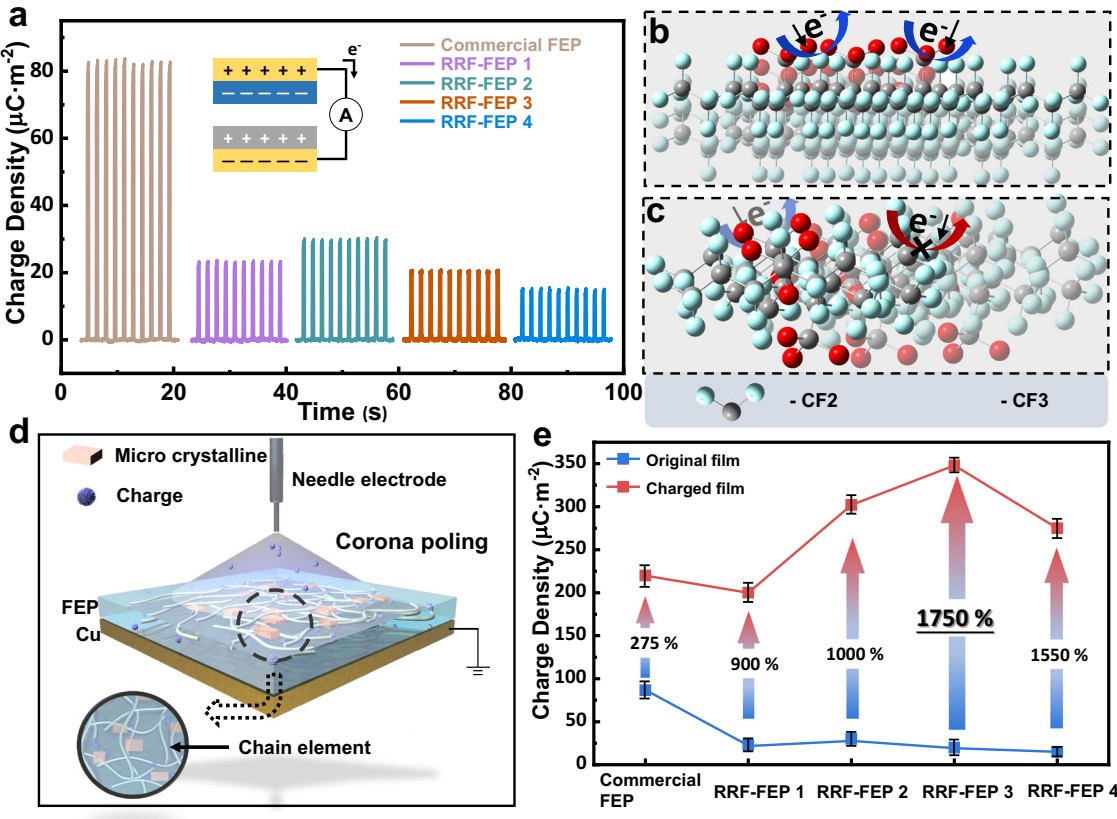

**Fig. 3 Effect of corona polarization on the performance of FEP. a** Maximum saturated charge density between FEP (without corona poling) and PA measured by contact-separation TENG. **b** The schematic models illustrating the commercial FEP. **c** The schematic models illustrating the RRF-FEP. **d** Schematic diagram of corona poling process. **e** Schematic diagram of charge injection process. **e** Comparison diagram of saturated charge density of FEP films before and after polarization. The error bar is obtained by repeating the test for ten times. Source data are provided as a Source Data file.

On the other hand, the saturated charge density of RRF-FEP after polarization has increased significantly, which can be explained by the difference in crystallinity. In Fig. 4c, the brightness of the diffraction ring also shows the intensity of the diffraction peak. Combined with the patterns of X-ray diffraction (XRD) (Fig. 4d), all samples show similar peak position, indicating that no crystal transition has occurred during RRF processing. Moreover, the crystallinity of each FEP is calculated and the change tendency of crystallization is shown in Supplementary Fig. S7, in which RRF-FEP3 has the highest crystallinity. In the molding process of polymer, applying different pressures to the melt polymer can change the shear flow induced polymer crystallization behavior. Specifically, the applied pressure can reduce the free volume of the polymer, increase the viscosity and impede the diffusion of molecular chain, all of which can finally affect its crystallization behavior. At the same time, the equilibrium melting point of the polymer and the undercooling level increases under pressure, which affects the nucleation process of the system[33]. Moreover, under the action of shear, the molecular orientation leads to the change of the free energy of the melt and further leads to the change of nucleus density of the crystallization process. In this case, the relaxation time ($t_{off}$) of molecules from a confined state to a thermodynamic equilibrium state is the key parameter to determine the crystallinity. By choosing an appropriate $t_{off}$ during the repeated forging process (over 120 times), the molecular structure of FEP can be set into a specially ordered form and the crystallization process can be enhanced during cooling. The $t_{off}$ has an optimized value for increasing the crystallinity, which is 2.1 s according to the

experimental results, and accordingly, the saturated charge density of RRF-FEP3 after polarization is also the highest. The influence of crystallinity on the improvement of triboelectrification capacity is mainly divided into two parts. First, high crystallinity means that the molecules are orderly arranged, resulting in a stronger polarization capability. Meanwhile, for FEP with higher crystallinity, a large number of deep traps form around the interface between microcrystalline and amorphous regions, which can also facilitate charge storage[34,35]. The induced electrons or ions are easier to be gathered at this interface, resulting in a stronger interface polarization and an enhanced charge storage capability[36-38]. The crystallinity of RRF-FEP3 has reached 58% and thus, its charge density can reach 352 μC·m⁻².

In addition, we measure the relative dielectric constant (ε) of these FEP films (see Supplementary Fig. S8). The ε of FEPs produced by RRF also increases significantly, which agrees with the result of saturated charge density. The increase of crystallinity also increases the dielectric constant of FEP films. The highest ε of 3.86 is with RRF-FEP3, which also has the highest crystallinity of 58%. Meanwhile, according to the Clausius–Mosotti equation[39], the increase of dielectric constant is due to the increase of FEP density induced by the dense packing in the RRF process. In addition, although the crystallinity of commercial FEP film is higher than that of RRF-FEP4, its saturated charge density after polarization is not as high as that of RRF-FEP4, which is due to the impurities of some O elements in the commercial film (see in Supplementary Fig. S9). The positron annihilation lifetime spectra[40] (PALS) of commercial film, RRF-FEP1 and RRF-FEP3 are characterized, in order to verify the packing states inside the

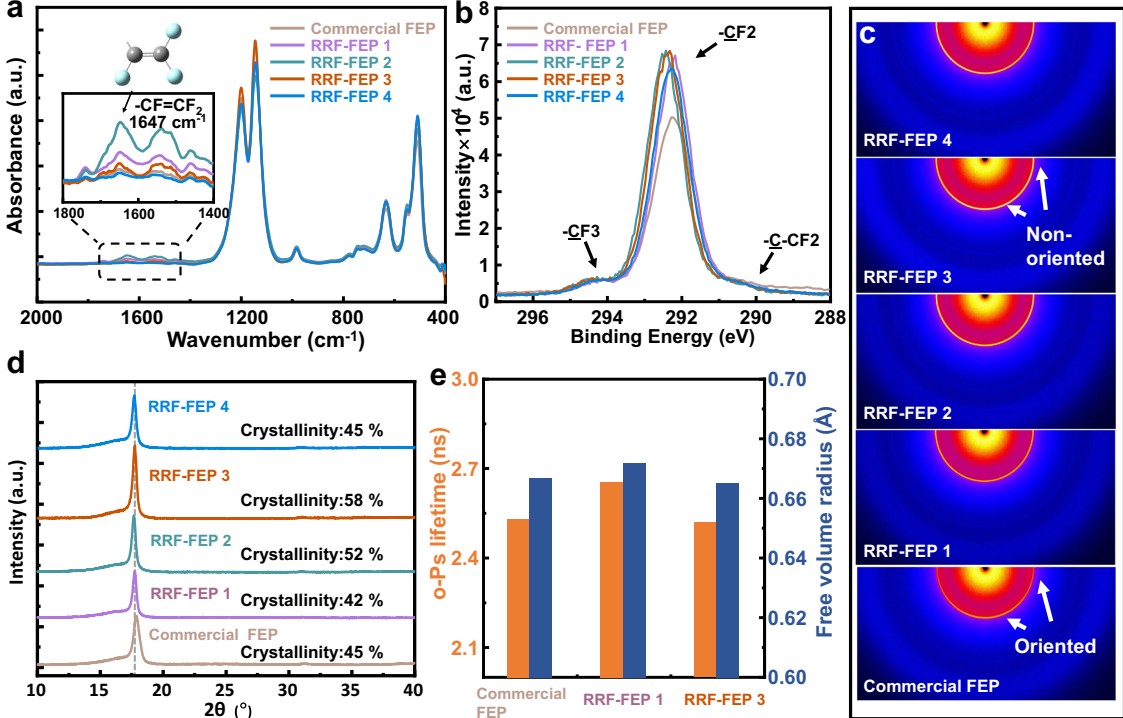

**Fig. 4 Characterization of FEP films. a** The molecular structure diagrams of several FEP films structures are given by a Fourier transform infrared (ATR-FTIR) spectrum. **b** C 1 s XPS spectra of FEP films. **c** 2D-WAXD patterns of FEP films. **d** X-ray diffraction (XRD) pattern of FEP films. **e** The o-Ps lifetime and free volume radius of FEP are calculated by positron lifetime spectrum. Source data are provided as a Source Data file.

films[41]. The PALS spectra were analyzed into three lifetime components[41] and the ortho-positronium (o-Ps) lifetime has a strong correlation with the size of the free volume. The free volume radius of the materials calculated by the formula in Table S4 is shown in Fig. 4e and there is no significant difference in the free volume of molecules between the three films. The intensity of o-Ps of the three films is very low, which shows that the compactness of the three films is very good. Then, it is worth noting that this is the free volume of the bulk overall region that may be different for the surface, which corresponds to the observations of $-CF_2$ by infrared and XPS.

**High performance of RRF-FEP film on TENG.** To further demonstrate the high performance of RRF-FEP film, the CS-TENG fabricated by RRF-FEP3 has been employed for testing its stability, power density and charging capacity. As shown in Fig. 5a, the RRF-FEP3 CS-TENG remains stable output after 1720 cycles in 2235 s, which shows the excellent charge capture and retention ability of RRF-FEP. Supplementary Fig. S10 shows the saturated charge density in contact with PA of the same corona polarized FEP film before and after being placed in the air for 48 h. The charge density decreased by only 7.4% after being placed for 48 h, which proved that the charge injected into FEP by corona polarization can be maintained for a considerable period. Figure 5b, c show the open-circuit voltage and peak power density of the CS-TENG at different resistances from 1 MΩ to 1 GΩ. The maximum peak power density of RRF-FEP3 CS-TENG reaches 150.1 mW·m$^{-2}$ with a load of 50 MΩ, and the maximum power density of commercial FEP CS-TENG reaches 89.7 mW·m$^{-2}$ with a load of 50 MΩ, which proves that the output power has been greatly improved. Using the circuit diagram shown in Supplementary Fig. S11, rectify the output of CS-TENG and charge the capacitor, and Fig. 5d–f compares the charging speed of RRF-FEP3 with commercial FEP, and more details are shown in Supplementary Movie 2. For RRF-FEP3, the

capacitors of 1 μF, 2.2 μF, 4.7 μF and 10 μF are charged to 5.5 V, 3.5 V, 2.0 V and 1.0 V within 200 s, respectively. It is important to note that CS-TENG is more suitable for sensory applications and it is not the optimized working mode for charging capacitors. Hence, we prepared a grating-structured freestanding TENG to demonstrate the charging capability. In this case, using RRF-FEP3 as the triboelectric layer, the capacitors of 10 μF are charged to 50 V within 9.2 s and the power density for charging can reach 720 mW·m$^{-2}$ (see Supplementary Fig. S12). Direct-current TENG (DC-TENG) based on air-breakdown is a recently developed TENG mode to couple the triboelectrification effect and electrostatic breakdown, whose working principle[4,5] is shown in Fig. 5g. Here, FEP is negatively charged and Cu is positively charged and the high voltage between the side electrode and FEP breaks through the air, leading to an electron transferring from the side electrode to Cu. This structure can effectively collect part of the energy wasted by air breakdown, so it can greatly improve the output of nanogenerator. As shown in Fig. 5h, i, using RRF-FEP3 on DC-TENG, up to 510 μC·m$^{-2}$ outputs are obtained, which is 1.2 times that of previously reported[4]. These results prove that the RRF-FEP film can benefit the study of TENG devices with diversified structures.

**Methods of improving triboelectric properties of polymers.** The triboelectrification ability of materials is affected by various parameters, including the functional groups, the orientation, the macroscopic crystallization and so on. Hence, we clarify the detailed contributions of these elements on the basis of current work. As shown in Fig. 6a, at the atomic level, the electronegativity of atoms determines the ability of electron capturing of functional groups, and the electron-withdrawing ability of these functional groups on the main chain can determine both the polarity and density of tribo-induced charges. In addition, at the chain level (see Fig. 6b), the orientation of the molecular chain affects the triboelectrification ability, since the orientation of the

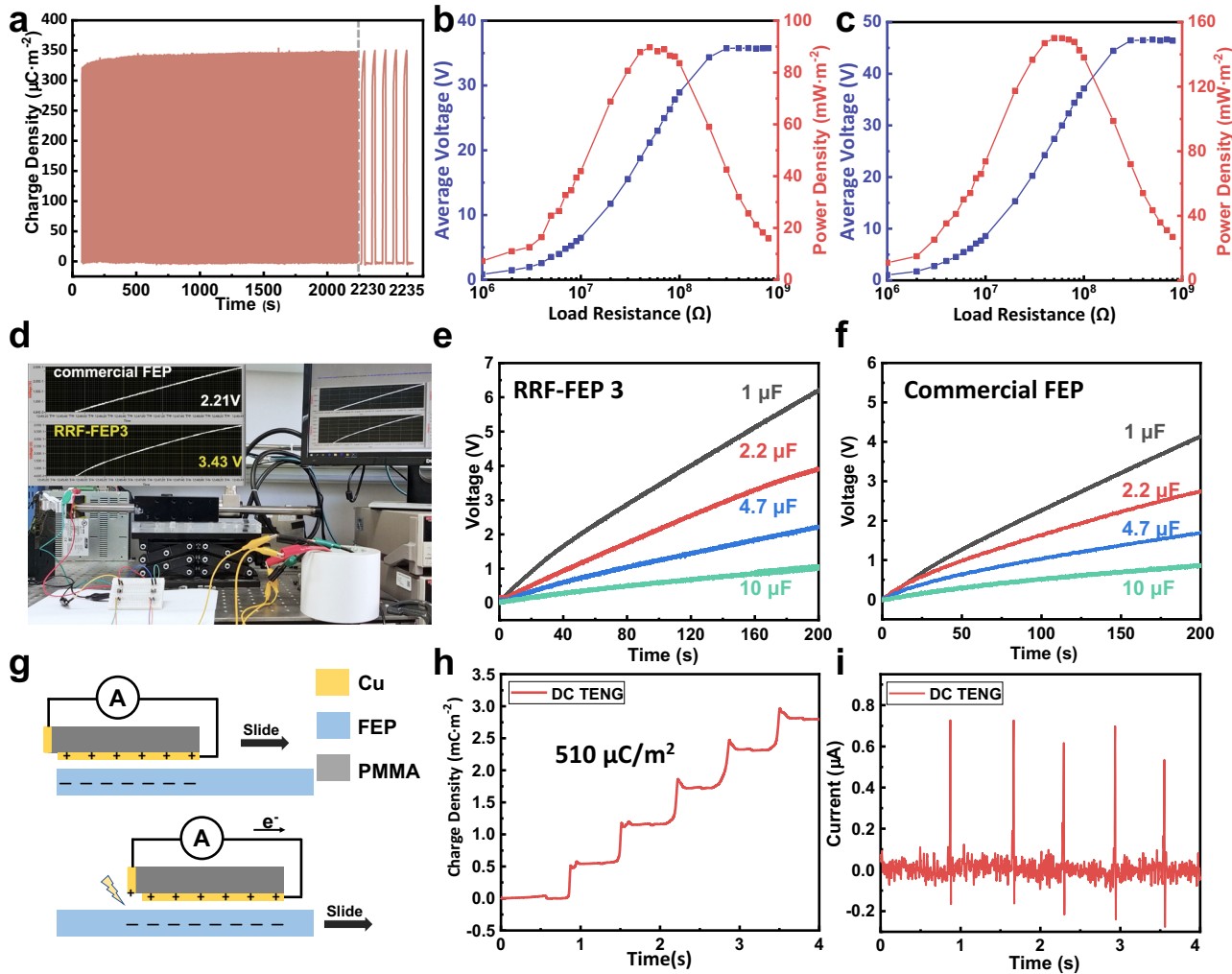

**Fig. 5 High charge density performance of RRF-FEP films. a** Stability test of the RRF-FEP3 CS-TENG with 2200 s. **b** Output current, voltage, and peak power density under various external loads of commercial FEP CS-TENG. **c** Output current, voltage, and peak power density under various external loads of RRF-FEP3 CS-TENG. **d** Comparison diagram of commercial FEP and RRF-FEP3 charging 2.2 μF capacitor at the same time. **e**, **f** Charging curve of 1, 2.2, 4.7 and 10 μF capacitor using the RRF-FEP3 CS-TENG. **g** Schematic diagram of working principle of DC-TENG. **h** Charge density and (**i**) Current of DC-TENG with RRF-FEP3. Source data are provided as a Source Data file.

molecular chain determines the functional groups exposed on the surface region, which have a higher possibility to actually contact with the counter material. Meanwhile, orientation also leads to the different densities of functional groups in the surface region, also affecting the probability of electron cloud overlap. At the level of bulk (see Fig. 6c), the crystallinity and the related deep traps caused by a large number of chains stacked affect the triboelectrification performance. The high crystallinity allows the molecules to be orderly arranged, resulting in larger dipole moments and higher dipole polarization Moreover, the interface between crystalline and amorphous regions leads to the formation of deep traps, which can also enhance the charge storage[35]. However, exceeding increase of crystallinity may also suppress the surface charge density. Since the interface is the key element for trapping and charge storage, the crystallinity may have an optimized value to cooperate with the charge injection process. It is important to note that these effects from the atomic level to the bulk level are synergistic and should be combined together for boosting the performance of triboelectric polymer. For example, for typical electret materials, including FEP (as shown in Supplementary Fig. S13), their saturated charge density is different when contacting with different materials (positive and negative),

which proves that functional groups still play a decisive role in contact electrification. As shown in Fig. 6d, the molecular structure of triboelectric polymers can regulate the electrical properties from the nano to the macro level. We expect to obtain a triboelectric material with negative functional groups, high density of surface functional groups and high crystallinity through a certain synthetic method. RRF is a good process to regulate these characteristics of materials, and to produce high friction electric materials. In this paper, the structure of the triboelectric polymer, including the functional groups and the bulk crystallinity, can be adjusted through RRF process, leading to the diversified change of its mechanical properties and electrification performances (Fig. 6e). Then, the related RRF process targeting the triboelectric materials with different requirements can be freely adjusted. Based on the achievement in this work, it is said that for non-electret materials, the selection of short $t_{off}$ may increase the number of end groups and give more chances to modify the functional groups in the surface region, which may effectively improve the triboelectric charge density. While, for the electret material, a longer $t_{off}$ can be selected to ensure the higher crystallinity and more deep traps can be generated in the bulk region.

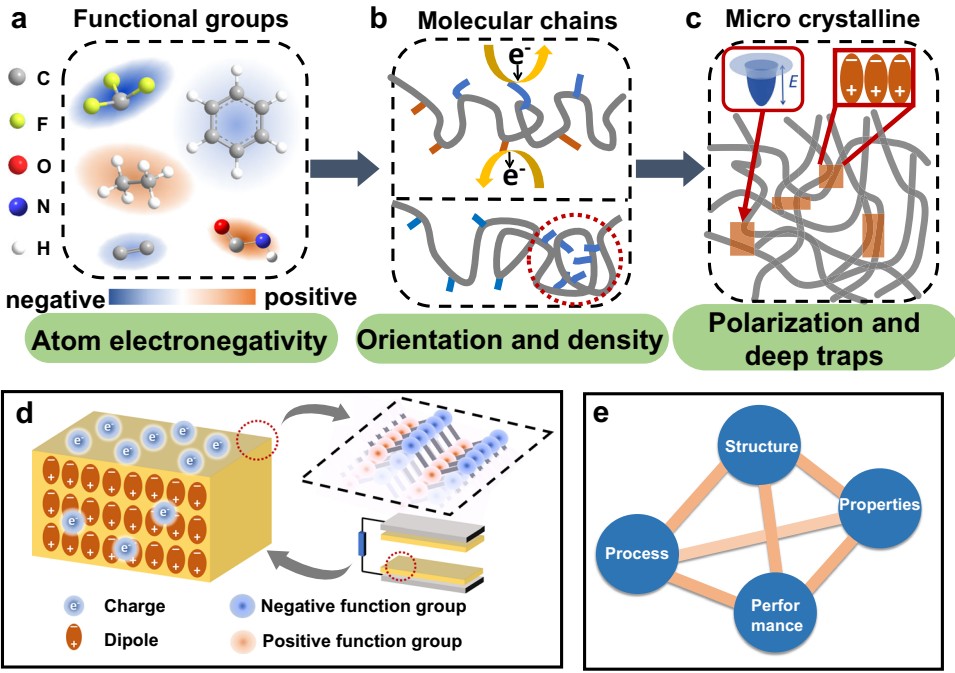

**Fig. 6 The key parameters for elevating the electrification performance.** Influence of **a** Functional groups (**b**) Molecular chains (**c**) Microcrystalline and **d** Synergistic effect on contact electrification ability of polymer. **e** The diagram of the polymer is successfully adjusted through a processing, so as to further change its properties and performances.

## Discussion

In this work, we have developed a molding method to control the molecular structure of triboelectric polymer by RRF process, which can be used to prepare polymer films with ultrahigh surface charge density. The FEP is selected as the target material for RRF fabrication, due to its ranking is an extreme negative position on the triboelectric series and the best commercial FEP tape obtained from the market is also tested to demonstrate the superior capability of RRF fabrication method. The produced FEP-RRF film can maintain excellent mechanical toughness and light transmittance, while its intrinsic surface charge density reaches $352\,\mu\text{C}\cdot\text{m}^{-2}$, which is 1.46 times of the highest records reported before. Through the systematic analysis, we find that the proposed RRF process can effectively regulate the composition of near surface functional groups the crystallinity and the dielectric constants of FEP films. By adjusting the changes in molecular structure and crystallinity, the distinct electrification performance before and after corona polarization can be observed, which can be used to respectively modify the triboelectric performance of electret and non-electret polymers. Due to its good mechanical toughness, light transmittance and ultra-high charge density, the prepared FEP-RRF film can be used for many occasions, such as energy collection, sensing and electrostatic adsorption. In the demonstration part, the fabricated FEP-RRF film has been applied to fabricate CS-TENG, freestanding-TENG and even the TENG with air-breakdown mode, while the ultrahigh output charge density can be achieved with all these working modes. Hence, the breakthrough achieved in materials development can benefit the study of TENG devices with various structures and applications. Moreover, we have summarized the key parameters for elevating the electrification performance on the basis of molecular structure and related fabrication crafts, which can guide the further development of triboelectric polymers. Accordingly, the RRF molding process is also expected to be used for various polymers to realize the controllable preparation of advanced triboelectric materials.

## Methods

**Materials**.

1. Commercial FEP: commercial FEP was bought from Shanghai Weitelang Industrial Co., Ltd.
2. RRF-FEP: Before processing, the FEP powders (9835, DuPont Co., Ltd) were stored in a vacuum oven at 60 °C for 6 h to remove the potential moisture. After then, the FEP powders were compression molded into 30–50 μm thickness sheet at 300 °C. To generate the RRF during processing, a custom-made compression molding machine was utilized with its programmable logic controlling system was modified to control the moving plate down and up for designed compressing and releasing. During processing, the pressure applied to polymer melt was recorded to monitor the pressure applied on the melt by dividing the pressure by the area of the mold. Four RRF conditions were chosen based on our pre-examination and were applied in this process, and the detailed parameters are shown in Supplementary Table S1, and the device is shown in Fig. S2 and we further prepared a video material (Supplementary Movie 1) to show the operation of the system. For every process, 120 repeats of unit were conducted to perform the samples based on our pre-examination. After molding, samples were cooled in air.

**Treatment before measurement**. Before the measurements, all the polymer films cleaned by absolute ethanol were cut into $10 \times 10\,\text{mm}^2$, and then placed in a vacuum drying oven for 3 h to remove all the initial charges. Samples requiring polarization are corona polarized at a voltage of 8 kV.

**Sample characterization**. A programmable electrometer (Keithley 6514) was used to test the open-circuit transferred charge, discharge current, and resistance. The digital linear motor was used to control the contact-separation process of measurement to ensure that the frequency and distance of TENG are the same. A scanning electron microscope (FEI Quanta FEG 250)was used to the elemental analysis. An atomic force microscope (Bruker Dimension Icon) was used to measure the surface morphology and roughness of films. A Fourier transform infrared spectroscope spectrometer (VERTEX80v, Bruker) was used to measure the structure of the elastic films. The materials were measured in attenuated total reflectance mode (ATR) was scanned from 4000 to 500 cm$^{-1}$. An X-ray photo-electron spectrometer (ESCALAB 250Xi) was used with an Al Kα excitation (1486.8 eV). An X-ray powder diffraction (Xpert3 Powder) and 2d-wide angle X-ray diffractometer (HomeLab) were all utilized to characterize the crystallinity and orientation of materials. Dielectric properties at the frequency of 1 kHz at room temperature were characterized by Novo Control Concept80. A Dynamic Thermomechanical Analyzer (TA DMA Q800) was used to measure the storage

modulus and loss angle of the materials in the range of 0–200 °C. (Tensile mold with frequency of 1 Hz, temperature ramping rate of 3 °C·min$^{-1}$ and amplitude of 5 μm).

## Data availability

The data supporting the findings of this study are reported in the main text or the Supplementary Information. Raw data are provided as a Source Data file. Source data are provided with this paper.

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

## Acknowledgements

This work was supported by the National Key R&D Project from Minister of Science and Technology (2021YFA1201601, 2019YFC1908202), the National Natural Science Foundation of China (62174014, 52103027), Beijing Nova program (Z201100006820063) and Youth Innovation Promotion Association CAS (2021165).

## Author contributions

X.C. and Z.-X.H. conceived the idea. Z.L.W. and J.-P.Q. supervised the experiment. Y.H., H.H. and Z.-X.H. fabricated all samples. Z.L. and X.C. prepared the manuscript. Z.L. and Y.S. designed the structure of the device. Z.L. performed the data measurements. F.C. performed PALS measurements. Y.S. and X.T. offered assistance with the experiments. All the authors discussed the results and commented on the manuscript.

## Competing interests

The authors declare no competing interests.
