## [Peer Review File · Nature Communications]

Fabrication of triboelectric polymer films via repeated rheological forging for ultrahigh surface charge densityREVIEWER COMMENTS

Reviewer #1 (Remarks to the Author):

In this paper, a high-performance fluorinated ethylene propylene film was synthesized by a molding method of repeated rheological forging. The synthesized FEP not only exhibited excellent triboelectric performance, but also significantly improved dielectric constant and mechanical properties, which can be used in various kinds of TENGs to improve their outputs. Moreover, this paper systematically summarizes the design strategies of triboelectric materials to provide instructive opinions for the future design of triboelectric materials. In summary, the proposed molding method in this article is quite interesting, and the achieved charge density may update a new record for TENG. The experiment is complete and it is of great value to improve the output of TENGs in the future. There are some minor suggestions for this work:

1. I note that in the article additional charge was injected into the FEP film by corona polarization, but it is not proven that this additional charge can be stored long term inside the material. The authors should demonstrate the preservation time of this portion of the charges to illustrate the use value of materials.
2. Deep trap was mentioned in fig.6a, and the authors mentioned that "a large number of deep traps form around the interface between microcrystalline and material, which can also facilitate charge storage", but there was no substantial evidence in the article, and I think that experiments by heat stimulated current are needed to prove the authors' statement.
3. The authors need to elaborate more clearly on the mechanism by which this molding process works leading to the enhancement of FEP crystallinity and end groups from the perspective of polymer rheology.
4. Moreover, the detailed preparation process of the repeated rheological forging is not clearly introduced. More detailed parameters related to the molding machine system should be provided.
5. Dielectric constant reflects the relative ability of a dielectric to store electrostatic energy in an electric field. The dielectric constant of FEP is only 2.1, and even the optimal sample RRF-FEP 6 in this paper has a dielectric constant of only 3.5, which is much lower than that of PVDF based polymers. However, the FEP produced in the article demonstrated superior storage charge capability, which the authors need to explain.
6. I think that the summary in Fig. 6 for the Design ideas for high performance triboelectric materials is interesting. However, the design strategy of functional groups, orientation and crystallinity to improve the properties of triboelectric materials has not been clearly described, and the schematic diagram also needs to be modified so that the results of the study are clear to the reader.

Reviewer #2 (Remarks to the Author):

In the manuscript, the authors introduced a molding method to fabricate polymer film with ultrahigh surface charge density. This process called RRF can achieve excellent mechanical toughness and light transmittance, while its intrinsic surface charge density reaches $352 \mu\text{C}\cdot\text{m}^{-2}$, which is 1.46 times of the highest records reported before. To prove this RRF works, they observe FEP-RRF roughness through AFM and SEM, and some DFT simulation data to compare FEP-RRF molecular structure changes. By adjusting the changes in molecular structure and crystallinity, the distinct electrification performance before and after corona polarization can be observed, which can be used to respectively modify the triboelectric performance of electret and non-electret polymers. Moreover, we have summarized the key parameters for elevating the electrification performance on the basis of molecular structure and related fabrication crafts, which can guide the further development of triboelectric polymers. Accordingly, the RRF molding process is also expected to be used for various polymers to realize the controllable preparation of advanced triboelectric materials. . Although, the authors summarized the RRF process and corona polarization well, there are some parts to be corrected.

The drawbacks and deficiencies are listed below.

Comment 1:

The authors make some process called RRF and I am curious about the forging power. Are there any correlation between the pressure(power) and flatness of the film surface?

Comment 2:

There are some typos in this article. On page 6, please change For Commercially available~ to For commercially available~. On supplementary information, I think the corresponding author was spelled wrong(Z.L W.).

Comment 3:

I cannot find any relation Figure S9 can explain typical electret material. It would be better explain more types of electret materials in table.

Based on the above comment, it seems novelty in triboelectric enhancing performance, and I recommend this manuscript for publication in Nature communications

Reviewer #3 (Remarks to the Author):

The authors used a method called repeated rheological forging to increase the intrinsic surface charge density of fluorinated ethylene propylene film. Repeated rheological forging involves the fluorinated ethylene propylene film molten at 300 oC, followed by repeating the fabrication process of pressing-releasing the film 120 times. This repeated rheological forging process with different releasing times is able to effectively regulate the molecular orientation, crystallinity and dielectric constant of the material. These changes result in distinct electrification performances before and after corona polarization. This suggests that different approaches could be used to boost the triboelectric performance of electret (fluorinated ethylene propylene after corona polarization) and non-electret (fluorinated ethylene propylene before corona polarization) polymers. For non-electret polymers, a shorter releasing time may increase the number of end groups. This would provide more chances to modify the functional groups in the surface region, which improves the triboelectric charge density. For electret polymers, a longer releasing time would ensure the higher crystallinity and allow more deep traps to be generated in the bulk region. This would improve the triboelectric charge density. I would recommend accepting the manuscript with minor revision.

Comments:

Could the authors provide some possible speculation on why the releasing time would have an optimal value, for having optimal amount of =CF₂?

Could the authors provide a more detailed speculation of why the crystallinity decreases when the releasing time increases beyond the optimal value?

If the performance of the film after repeated rheological forging only improves over that of the original commercial fluorinated ethylene propylene film, with high voltage source used to perform corona polarization, would this limit the ease of operation?

Could the authors provide a clearer explanation for the fundamental mechanism of improving the surface charge density, such as how the factor of crystallinity is fundamentally related to the increase in polarizability, leading to higher charge density?

Reviewer #4 (Remarks to the Author):

This manuscript reports a process that allows to generate triboelectric films with ultrahigh charge densities. The properties of the films seem promising. There are however a number of concerns with regards to the materials characterization (or need thereof) that need to be clarified and improved.

1) In lines 91 - 93, the authors state that under pressure molecular chains are compressed.... and there is breaking of chains. And after that the chain reach their relaxed state. The author need to rephrase this. This is not the wording one would use in a polymer science paper. There is also no experimental evidence in the paper that convincingly demonstrates that chains are cleaved. What do the authors mean when they say that chains recover to reach their "relaxed" state. What is that and can that be measured ?

2) lines 116 - 118. The statement on the effect on forging and that it "makes the chains in a highly condensed state" is not a proper description to describe polymer structure and behavior. Can the authors do e.g. dynamic mechanical experiments to assess the effects of forging on mechanical properties ?

3) Lines 172 - 173: the authors claim the presence of C=C bonds. Where do these come from. As far as this review is aware there are no C=C bonds present in the polymer.

4) Lines 198 - 214 discuss the crystallinity of polymer samples subjected to different numbers of forging cycles. In Figure 4C crystallinities are given with accuracies of two digits after the comma. Is this level of precision correct ? What is the error / uncertainty in the determination of the crystallinity of the polymers ? It seems a little surprising that there is no clear trend in the reported crystallinities, and it raises the question whether the reported differences are statistically significant.

5) Can the authors perform TGA analysis on the pristine polymer, as well as on samples that have been subjected to different forging cycles ? Maybe this can help to provide insight into possible thermal degradation of the polymer as it is subjected to an increasing number of forging cycles ?

Reviewer #1

In this paper, a high-performance fluorinated ethylene propylene film was synthesized by a molding method of repeated rheological forging. The synthesized FEP not only exhibited excellent triboelectric performance, but also significantly improved dielectric constant and mechanical properties, which can be used in various kinds of TENGs to improve their outputs. Moreover, this paper systematically summarizes the design strategies of triboelectric materials to provide instructive opinions for the future design of triboelectric materials. In summary, the proposed molding method in this article is quite interesting, and the achieved charge density may update a new record for TENG. The experiment is complete and it is of great value to improve the output of TENGs in the future. There are some minor suggestions for this work:

Comment 1: I note that in the article additional charge was injected into the FEP film by corona polarization, but it is not proven that this additional charge can be stored long term inside the material. The authors should demonstrate the preservation time of this portion of the charges to illustrate the use value of materials.

Reply1: Thank you for your comments. Many previous studies have proved that the charge injected into FEP film through corona polarization can be stored for a long time (IEEE Transactions on Dielectrics and Electrical Insulation 10(1):102 - 108). As for TENG, in our experiment, we have measured the output of RRF-FEP3 CS-TENG with 1720 cycles in 2235s, which shows the excellent charge capture and retention ability of RRF-FEP. Moreover, we additionally tested the saturated charge density in contact with PA of the same corona polarized FEP film before and after being placed in air for 48 hours (Figure S10). It was found that the charge decreased by only 7.4% after being placed for 48 hours, which proved that the charge injected into FEP by corona polarization can be maintained for a considerable period. In the process of actual use, since the contact electrification process will continuously supplement the charge to the material surface, this part of the charge will exist for a longer time without re corona polarization. In general, this TENG has a strong ability to maintain the polarization charge in practical application. We further explained this part, as can be seen in page11 and Supplementary Figure S10.

Figure R1. Charge density of PA-FEP after corona poling and after placing in the air for 48h.

Comment 2: Deep trap was mentioned in fig.6a, and the authors mentioned that “a large number of deep traps form around the interface between microcrystalline and material, which can also facilitate charge storage”, but there was no substantial evidence in the article, and I think that experiments by heat stimulated current are needed to prove the authors' statement.

Reply2: Thank you for your comments. The contributions of deep traps to the changing density as well as the stability of charges have been well explained in our previous work [Adv. Funct. Mater. 31 (49), 2106082]. Instead of heat stimulated current, the atomic force microscopy measurement can clearly identify the concentration of electric field near the interface, where a large amount of deep traps are filled. We further explained this part, as can be seen in page10 and 12.

Comment 3: The authors need to elaborate more clearly on the mechanism by which this molding process works leading to the enhancement of FEP crystallinity and end groups from the perspective of polymer rheology.

Reply 3: Thank you for your comments. In the molding process of polymer, applying pressure to the melt polymer will lead to the change of flow induced polymer crystallization behavior. As presented in our manuscript, the RRF can be divided into two stages: pressing section and releasing. In the compression step, pressure was applied on the confined polymers, which induced decreased section free volume and enhanced order level. While in the release step, no pressure was applied, and the chains would recover from the highly confined thermodynamically unfavorable state back to a disordered and entanglement state. Specifically, the pressure will reduce the free volume of the polymer melt, increase the viscosity of the system, increase the difficulty of molecular chain diffusion, forcing the chains to be ordered, and affect its crystallization behavior. At the same time, the equilibrium melting point of the polymer increases and the undercooling increases under pressure, which affects the nucleation process of the system. (Macromolecules 1983, 16, 1, 55–59; Macromolecules 2000, 33, 4138-4145). RRF molding process applies dynamic pressure field to the melt polymer, which changes the crystallization behavior of the polymer. Moreover, we think that the relaxation time of molecules from a thermodynamic unfavorable state back to an equilibrium one is the key to the formation of polymers with different structures, and t_{off} should have an optimal match with the time of molecular relaxation process, which is the reason for the best t_{off} . We further explained this part, as can be seen in page 9-10.

Comment 4: Moreover, the detailed preparation process of the repeated rheological forging is not clearly introduced. More detailed parameters related to the molding machine system should be provided.

Reply 4: Thank you for your comments. At your suggestion, we explained the process of RRF process in more detail, and added a video to show the molding process. Before processing, the FEP powders (9835, DuPont Co., Ltd) were stored in a vacuum oven at 60 °C for 6 h to remove the potential moisture. After then, the FEP powders were compression molded into 30-50 μ m thickness sheet at 300 °C. To generate the RRF during processing, a custom-made compression molding machine was utilized with its PLC controlling system was modified to control the move plate down and up for designed compressing and releasing. During processing, the pressure

applied on polymer melt was recorded to monitor the pressure applied on the melt by dividing the pressure by the area of the mold. Four RRF conditions were chosen based on our pre-examination and were applied in this process, and the detailed parameters are shown in Supplementary Table S1, and the device is shown in Figure S2 and we further prepared a video material (Supplementary Video S1) to show the operation of the system (the screenshot is shown in Figure R2). For every process, 120 repeats of unit were conducted to perform the samples based on our pre-examination. After molding, samples were cooled in air. We further explained this part, as can be seen in page15 and Supplementary Video S1.

Figure R1. The demonstration of the repeated rheological forging (RRF) process in this article.

Comment 5: Dielectric constant reflects the relative ability of a dielectric to store electrostatic energy in an electric field. The dielectric constant of FEP is only 2.1, and even the optimal sample RRF-FEP 6 in this paper has a dielectric constant of only 3.5, which is much lower than that of PVDF based polymers. However, the FEP produced in the article demonstrated superior storage charge capability, which the authors need to explain.

Reply5: As for the influence of dielectric constant on the output performance of TENG, it has been proved that increasing the relative dielectric constant of materials by doping and other methods can improve the triboelectric properties of materials (Adv. Mater., 27: 4938-4944.). However, in our previous study (Adv. Mater. 2020, 32, 2001307.), we found that the triboelectrification ability is also related to the electronic gain and loss ability of functional groups. For molecules with a higher proportion of -F groups, the triboelectrification ability will become stronger. Moreover, for the materials with high dielectric constant, the large amount of charge stored in ferroelectric material increases the repulsion ability of charge. Accordingly, its shielding ability of charge also increases, so it is more difficult to contain new charge. We further explained this part, as can be seen in page12-13.

Comment 6: I think that the summary in Fig. 6 for the Design ideas for high performance triboelectric materials is interesting. However, the design strategy of functional groups, orientation and crystallinity to improve the properties of triboelectric materials has not been clearly described, and the schematic diagram also needs to be modified so that the results of the study are clear to the reader.

Reply6: We thank you for your construction suggestion. According to your suggestion, we have modified the content of Fig. 6, in order to provide a better explanation for this point. As shown in Fig.6, combined our previous studies, we summarized the key elements in the preparation of high-performance triboelectric materials from the perspective of molecular design. We further explained this part, as can be seen in page12-13.

Figure 6. The key parameters for elevating the electrification performance. influence of a) functional groups b) molecular chains c) micro crystalline and d) synergistic effect on contact electrification ability of polymer. e) The diagram of the polymer is successfully adjusted through a new processing, so as to further change its properties and performances

Reviewer #2

In the manuscript, the authors introduced a molding method to fabricate polymer film with ultrahigh surface charge density. This process called RRF can achieve excellent mechanical toughness and light transmittance, while its intrinsic surface charge density reaches $352 \mu\text{C}\cdot\text{m}^{-2}$, which is 1.46 times of the highest records reported before. To prove this RRF works, they observe FEP-RRF roughness through AFM and SEM, and some DFT simulation data to compare FEP-RRF molecular structure changes. By adjusting the changes in molecular structure and crystallinity, the distinct electrification performance before and after corona polarization can be observed, which can be used to respectively modify the triboelectric performance of electret and non-electret polymers. Moreover, we have summarized the key parameters for elevating the electrification performance on the basis of molecular structure and related fabrication crafts, which can guide the further development of triboelectric polymers. Accordingly, the RRF molding process is also expected to be used for various polymers to realize the controllable preparation of advanced triboelectric materials. Although, the authors summarized the RRF process and corona polarization well, there are some parts to be corrected.

Based on the above comment, it seems novelty in triboelectric enhancing performance, and I recommend this manuscript for publication in Nature communications

The drawbacks and deficiencies are listed below:

Comment 1:

The authors make some process called RRF and I am curious about the forging power. Are there any correlation between the pressure (power) and flatness of the film surface?

Reply1: Thank you for your comments. As shown in Figure 2a, we used Atomic Force Microscope (AFM) to measure the surface roughness of several RRF-FEPs, and they all molded by repeated rheological forging under the pressure of 50 MPa. The results show that these films are very flat, and the root mean square roughness is about 14 nm. Moreover, according to previous references about nano printing (Nano Lett. 2004, 4, 4, 633–637; Jpn. J. Appl. Phys. 49 (2010) 06GL01), when the pressure exceeds a certain threshold, the roughness is only decided by the morphology of template surface. In our experiment, we chose the pressure of 50MPa, and under this pressure, the surface morphology of the RRF film remains almost the same. It is worth noting that if the applied pressure is very low, the flatness may have some differences. However, based on our system, the pressure provided by our machine is usually larger than the pressure threshold of the imprinting and the influence of pressure to the surface roughness is negligible. We further prepared a video material (Supplementary Video S1) to show the operation of the system (the screenshot is shown in Figure R2).

Figure R2. The demonstration of the repeated rheological forging (RRF) process in this article.

Comment 2:

There are some typos in this article. On page 6, please change For Commercially available~ to For commercially available~. On supplementary information, I think the corresponding author was spelled wrong(Z.L W.).

Reply2: Thank you for your comments. We carefully checked the whole article and corrected many spelling mistakes, including those you mentioned. We have to mention that Prof. Z. L. W. is not the corresponding author in this work, since he is willing to support his student, Dr. X. C. to have independent researches and collaborations with other scholars. Thank you again for your suggestions on improving the overall quality of our article.

Comment 3:

I cannot find any relation Figure S9 can explain typical electret material. It would be better explain more types of electret materials in table.

Reply 3: Thank you for your suggestions. We further studied several common polymer electrets, including polytetrafluoroethylene (PTFE), polypropylene (PP) and polyvinylidene fluoride (PVDF), which can help to clarify the influence of functional groups on their triboelectrification performance cannot be ignored. As can be seen in Figure S13, the highest surface charge density is achieved with FEP film among all the electret materials, which explains why we choose FEP film to demonstrate the forging method. Meanwhile, the similar mechanism of the combined effects of surface functional groups and polarization can be observed with all the electret material. For the fully polarized FEP, PTFE, PP and PVDF film, their saturated charge density is different when contacting with PA and PET, Hence, even though the polarization can enhance the stored charges inside the film, functional groups still play a decisive role in contact electrification. Then, we can conclude that the polarization induced by external field can enhance the charge density of triboelectric polymer, but the functional groups can also decide the charge maintenance during the contact motion. The combination of two effects can finally determine the operation of TENG. We further explained this part, as can be seen in page12 and Supplementary Figure S13.

Figure S13. Charge density of electret a) FEP b) PTFE c) PP d) PVDF contact with PET and PA films.

Reviewer #3

The authors used a method called repeated rheological forging to increase the intrinsic surface charge density of fluorinated ethylene propylene film. Repeated rheological forging involves the fluorinated ethylene propylene film molten at 300 °C, followed by repeating the fabrication process of pressing-releasing the film 120 times. This repeated rheological forging process with different releasing times is able to effectively regulate the molecular orientation, crystallinity and dielectric constant of the material. These changes result in distinct electrification performances before and after corona polarization. This suggests that different approaches could be used to boost the triboelectric performance of electret (fluorinated ethylene propylene after corona polarization) and non-electret (fluorinated ethylene propylene before corona polarization) polymers. For non-electret polymers, a shorter releasing time may increase the number of end groups. This would provide more chances to modify the functional groups in the surface region, which improves the triboelectric charge density. For electret polymers, a longer releasing time would ensure the higher crystallinity and allow more deep traps to be generated in the bulk region. This would improve the triboelectric charge density. I would recommend accepting the manuscript with minor revision.

Comment 1:

Could the authors provide some possible speculation on why the releasing time would have an optimal value, for having optimal amount of =CF₂?

Reply 1: We thank you for your comments. This is a very good question that we have been considered for a long time. This RRF process can actually be divided into two parts: pressing section and releasing section. In the pressing section, the molecular chains of FEP are confined and compressed in the mold, resulting in the decrease of the free volume, and consequently induces ordered chain packing. Moreover, in this process, some level of cleavages would occur in the long chains, due to the overhigh compression. Then, in the releasing section, the compressed material starts to recover from such a thermodynamically unfavorable state back to a thermodynamically stable state, in which the chains tend to be disordered and entangled. If the relaxation time for the forging is long enough, the polymer molecules can all reach the thermodynamic equilibrium state. Meanwhile, if we apply different relaxation time to the polymer during the hot pressing, the polymer molecules can be fixed at a specially confined state. This relaxation time is the key to the formation of polymers with different structures. Accordingly, if the molecular structure can have an optimized states for electrification, there must be an optimized t_{off} to match with the state of molecular relaxation process, because if t_{off} is infinite long, this method has the same effect as ordinary hot-pressing molding, so the relaxation process in repeated forging is the key factor in forming. As for the formation of -CF=CF₂ group, the experimental results are confirmed by both Fourier transform infrared (FTIR) and X-ray photoelectron spectroscopy (XPS) measurement (see Figure 4a,b and Supplementary Figure S5). Usually, the -CF=CF₂ group is the end part of the molecular chain. Thus, we infer that, at least in the surface region, a certain amount of polymer chains is possibly cleaved during the repeated pressing process and a series of short chains are generated, resulting in more end groups (-CF=CF₂) observed in the experiments. We further explained this part, as can be seen in page 4,5 and 8.

Comment 2:

Could the authors provide a more detailed speculation of why the crystallinity decreases when the releasing time increases beyond the optimal value?

Reply 2: We thank you for your constructive comments. In the molding process of polymer, applying different pressures to the melt polymer can change the shear flow induced polymer crystallization behavior. Specifically, the applied pressure can reduce the free volume of the polymer, increase the viscosity and impede the diffusion of molecular chain, all of which can finally affect its crystallization behavior. At the same time, the equilibrium melting point of the polymer and the undercooling level increases under pressure, which affects the nucleation process of the system. (Macromolecules 1983, 16, 1, 55–59; Macromolecules 2000, 33, 4138-4145). Moreover, under the action of shear, the molecular orientation leads to the change of the free energy of the melt and further leads to the change of nucleus density of the crystallization process. In our work, the rheological forging applies dynamic field to the melt polymer, which changes the crystallization behavior of the polymer. Hence, the time ratio of this dynamic flow field is the key to affect the crystallization nucleation, and the relaxation time of molecules from a confined state to a thermodynamic equilibrium state is the key parameter for the formation of polymers with different structures, and t_{off} should have an optimal match with the time of molecular relaxation process. It is important to note that the high crystallinity is achieved at a confined state of molecular chain. In this case, if the releasing time exceedingly increases beyond the optimal value, the confined molecular chain may return to the full thermodynamic equilibrium state of common hot-pressing treatment, leading to the decrease the crystallinity. We further explained this part, as can be seen in page4,5 and 9.

Comment 3:

If the performance of the film after repeated rheological forging only improves over that of the original commercial fluorinated ethylene propylene film, with high voltage source used to perform corona polarization, would this limit the ease of operation?

Reply 3: We thank you for your constructive comments. In order to improve the output power of TENG, most FEP films are polarized before use, and this part of the charge can be maintained for a long time. In the real application, the supplement the charge induced on the material surface by corona polarization can exist for a very long time, which has been proved by many previous works (IEEE Transactions on Dielectrics and Electrical Insulation 10(1):102 - 108). In addition, we found that negative ions can be directly implanted into the surface of FEP films by ionizing air with electrostatic eliminator (Milty Zerostat3), and this simple operation can achieve the same effect as corona polarization. Hence, we have several easily accessible methods to achieve the stable polarization of FEP film, which can fully support the practical operation of this film.

Comment 4:

Could the authors provide a clearer explanation for the fundamental mechanism of improving the surface charge density, such as how the factor of crystallinity is fundamentally related to the increase in polarizability, leading to higher charge density?

Reply 4: Thank you for your constructive suggestion. The influence of crystallinity on the improvement of triboelectrification capacity is mainly divided into two parts. First, high

crystallinity means that the molecules are orderly arranged, resulting in a stronger polarization capability. Crystalline polarization can also be expressed as the large sample limit of the dipole of a bounded crystallite over its volume. Under the corona polarization of high-voltage electric field, the film with higher crystallinity producing more orderly areas, which makes it easier to produce dipole polarization (Electrical properties of polymers). In addition, the space charge of electret (including surface charge and volume charge) is stored in the trap energy level, which is located in the gap between conduction band and valence band. For FEP with higher crystallinity, a large number of deep traps form around the interface between microcrystalline and material, which can also facilitate charge storage. (IEEE Transactions on Dielectrics and Electrical Insulation 11, 739-753 (2004)) On the interface of crystalline and amorphous regions, under the action of external electric field, electrons or ions in the dielectric are easier to be gathered at this interface, resulting in a stronger interface polarization and an enhanced charge storage capability.

In addition to the influence of crystallinity on the contact electrification capacity of materials, the electron withdrawing capability of functional groups also has great influence on triboelectrification. For the fully polarized FEP film, their stabilized charge density is different by contacting with PA and PET (Figure S13), suggesting that polarization induced charge may be taken away during the contact motion. Hence, the polarization induced by external field can enhance the charge density of triboelectric polymer, but the functional groups can help to maintain the charge density during the contact. The combination of two effects can finally determine the operation of TENG. In order to address this comment, we have further modified the content of Figure 6, in order to provide a clearer explanation for this fundamental mechanism. We further explained this part, as can be seen in page12-13, where the key elements in the preparation of high-performance triboelectric materials from the perspective of molecular design have been summarized.

Figure 6. The key parameters for elevating the electrification performance. influence of a)

functional groups b) molecular chains c) micro crystalline and d) synergistic effect on contact electrification ability of polymer. e) The diagram of the polymer is successfully adjusted through a new processing, so as to further change its properties and performances

Reviewer #4

This manuscript reports a process that allows to generate triboelectric films with ultrahigh charge densities. The properties of the films seem promising. There are however a number of concerns with regards to the materials characterization (or need thereof) that need to be clarified and improved.

Comment 1: In lines 91 - 93, the authors state that under pressure molecular chains are compressed.... and there is breaking of chains. And after that the chain reach their relaxed state. The author need to rephrase this. This is not the wording one would use in a polymer science paper. There is also no experimental evidence in the paper that convincingly demonstrates that chains are cleaved. What do the authors mean when they say that chains recover to reach their "relaxed" state. What is that and can that be measured?

Reply 1: We sincerely acknowledge the reviewer for the careful corrections. We are very sorry that several misleading words were used in the initial draft, which were corrected in this revision. Besides, the reviewer raised concern around the cleavage of chains we demonstrated. Such statement was based on the observation of improved content of unsaturated group of $-CF=CF_2$, which is confirmed by both the Fourier transform infrared (FTIR) and X-ray photoelectron spectroscopy (XPS) measurement (see Figure 4a and 4b). Usually, the $-CF=CF_2$ group is the end part of the molecular chain. Thus, we infer that, at least in the surface region, a certain amount of polymer chains is possibly cleaved during the RRF process and a series of short chains are generated, resulting in more end groups ($-CF=CF_2$) observed in the experiments. The cleavage of molecular chain is possibly caused by the overhigh accumulated compression energy during rheological forging.

Moreover, in the sentence "chains recover to reach their "relaxed" state", we hope to say that under compression, the chains are confined in the mold room, which is a thermodynamically unfavorable state. Thus, when the pressure removed, the chains can relax from confined state back to the thermodynamically stable one, in which the chains are disordered and entangled. Meanwhile, the relaxation of molecular chain until the thermodynamically stable state also leads to the change of molecular structure as well as the crystallinity, which can all be measured by using FTIR, XPS and XRD (Figure 4a and 4b).

The corresponding sentence was updated as:

In the pressing section, the molecular chains of FEP are confined and compressed in the mold, resulting in the decrease of the free volume, and consequently induces ordered chain packing. Moreover, in this process, some level of cleavages would occur in the long chains, due to the overhigh compression. Then, in the releasing section, the compressed material starts to recover from such a thermodynamically unfavorable state back to a thermodynamically stable state, in which the chains tend to be disordered and entangled.

We further explained this part, as can be seen in page 4.

Comment 2: lines 116 - 118. The statement on the effect on forging and that it "makes the chains in a highly condensed state" is not a proper description to describe polymer structure and behavior. Can the authors do e.g. dynamic mechanical experiments to assess the effects of forging on mechanical properties?

Reply 2: We appreciate your comments. Again, we apologize for the improper language in

previous submission. In this sentence, we meant that owing to the closed volume of mold, as well as the compression applied, the polymer chains are in a highly confined state. The related sentence was corrected as: Makes the chains in a highly confined state.

We further explained this part, as can be seen in page 5.

In addition, as suggested by the reviewer, dynamic mechanical analysis (DMA) was conducted using a TA Q800 in tensile mold with frequency of 1 Hz, temperature ramping rate of 3 °C·min⁻¹ and amplitude of 5 μm to provide more information of the impact of RRF on the mechanical properties of FEP films in comparison with commercial ones. From the DMA data (see Figure S3), we can see that the molding method did not change the glass transition temperature of FEP, but in the entire temperature ranges from room temperature to 220 °C, our RRF FEP shows higher storage modulus, indicating a higher strength. Such observation agrees well with the tensile properties we reported in the manuscript. Since the storage modulus of crystalline phase is higher than that of amorphous phase, this result can also indirectly prove that RRF process does increase the crystallinity of the FEP film.

Fig. S3: The storage modulus and Tan delta of commercial FEP and RRF-FEP3 measured by dynamic mechanical analyzer.

Comment 3: Lines 172 - 173: the authors claim the presence of C=C bonds. Where do these come from? As far as this review is aware there are no C=C bonds present in the polymer.

Reply 3: We are very sorry that we used the wrong expression “C=C” for “-CF=CF₂”. As for the formation of -CF=CF₂ group, the experimental results are confirmed by both Fourier transform infrared (FTIR) and X-ray photoelectron spectroscopy (XPS) measurement (see Figure 4a and 4b). Usually, the -CF=CF₂ group is the end part of the molecular chain (Macromol. Mater. Eng., 291: 937-943). Thus, we infer that, at least in the surface region, a certain amount of polymer chains is possibly cleaved during the repeated pressing process and a series of short chains are generated, resulting in more end groups (-CF=CF₂) observed in the experiments. We further explained this part, as can be seen in page 8.

Comment 4: Lines 198 - 214 discuss the crystallinity of polymer samples subjected to different numbers of forging cycles. In Figure 4C crystallinities are given with accuracies of two digits after the comma. Is this level of precision correct? What is the error / uncertainty in the determination of the crystallinity of the polymers? It seems a little surprising that there is no clear trend in the reported crystallinities, and it raises the question whether the reported differences are statistically significant.

Reply 4: Thanks for your constructive comment for enhancing the quality of our work. The crystallinity of each sample was calculated through XRD pattern as

$$\chi = \frac{A_{cry}}{A_{total}}$$

, where χ is the crystallinity, A_{cry} and A_{total} are the area of crystalline peaks and total peaks that obtained from XRD, respectively. It is widely accepted that XRD is a precise method for evaluating the crystallinity of polymers. In our manuscript, the crystallinity is directly calculated based on a software (MDI Jade), where the accuracies of two digits after the comma are automatically output. However, it is important to note that the error bar of calculation from the software is larger than 1 %. In this case, the accuracies of two digits after the comma is not a good way to illustrate the crystallinity. We really appreciate the reviewer for pointing out this problem. The calculated crystallinity as well as the error bar are shown in Table R1, while we report the rounding value of crystallinity in the manuscript.

In addition, for clearly observation of the trend in crystallinities, we plotted the crystallinities as below (Figure S7). From which, we can see that the crystallinity of RRF FEP film increase and then decrease with longer releasing time used in the RRF technique. Suggesting that a proper RRF condition can largely enhance the crystallization behavior of FEP chains.

	output data	rounding data
Commercial FEP	45.46% \pm 1.75%	45%
RRF-FEP1	41.52% \pm 1.81%	42%
RRF-FEP2	52.39% \pm 1.4%	52%
RRF-FEP3	57.84% \pm 1.7%	58%
RRF-FEP4	44.85% \pm 0.76%	45%

Table R1: The crystallinity of output data and rounding data.

Fig. S7: The variation trend of crystallinity of RRF-FEPs.

Comment 5: Can the authors perform TGA analysis on the pristine polymer, as well as on samples that have been subjected to different forging cycles? Maybe this can help to provide insight into possible thermal degradation of the polymer as it is subjected to an increasing number of forging cycles?

Reply 5: Thanks for the valuable comment for helping us gain deeper insight of RRF. TGA of commercial FEP, raw FEP powders (0 cycle) and RRF-FEP3 (60 cycles, 120cycles and 180cycles) was provided as below (Supplementary Figure S4). The results revealed that all samples show similar thermal degradation behaviors with temperature from 25 °C to 800 °C. Comparing with the raw FEP powders (0 cycle), we can see that RRF FEP with more cycles show similar degradation temperatures, which indicates no obvious chain degradation occurred during the repeated forging process.

Fig. S4: Thermogravimetric (TGA) analysis diagram of commercial FEP and RRF-FEP3 with different RRF cycles.

REVIEWERS' COMMENTS

Reviewer #1 (Remarks to the Author):

The paper has been well revised based on the reviewers' comments and well polished. The reviewer believes it can be published in Nature Comm. Congratulations to the nice work.

Reviewer #3 (Remarks to the Author):

The authors have addressed my comments in this revised version of the manuscript. I recommend the publication of this manuscript.

Reviewer #4 (Remarks to the Author):

As I mentioned in the previous round of reviews, I can only comment on the polymer part as I am not really a tribo electric expert.

The authors in their revision however have satisfactorily addressed my concern, and for my part, I would be happy to recommend publication.

Reviewer #1:

The paper has been well revised based on the reviewers' comments and well polished. The reviewer believes it can be published in Nature Comm. Congratulations to the nice work.

Reply: Thank you very much for your support on our paper.

Reviewer #3:

The authors have addressed my comments in this revised version of the manuscript. I recommend the publication of this manuscript.

Reply: Thank you very much for your support on our paper.

Reviewer #4:

As I mentioned in the previous round of reviews, I can only comment on the polymer part as I am not really a tribo electric expert.

The authors in their revision however have satisfactorily addressed my concern, and for my part, I would be happy to recommend publication.

Reply: Thank you very much for your support on our paper.